# Effect of Expression of Human Glucosylceramidase 2 Isoforms on Lipid Profiles in COS-7 Cells

**DOI:** 10.3390/metabo10120488

**Published:** 2020-11-27

**Authors:** Peeranat Jatooratthawichot, Chutima Talabnin, Lukana Ngiwsara, Yepy Hardi Rustam, Jisnuson Svasti, Gavin E. Reid, James R. Ketudat Cairns

**Affiliations:** 1School of Chemistry, Institute of Science, Suranaree University of Technology, Nakhon Ratchasima 30000, Thailand; pj.cbsfa@gmail.com (P.J.); chutima.sub@sut.ac.th (C.T.); 2Center for Biomolecular Structure, Function and Application, Suranaree University of Technology, Nakhon Ratchasima 3000, Thailand; 3Laboratory of Biochemistry, Chulabhorn Research Institute, Bangkok 10210, Thailand; lukana@cri.or.th (L.N.); jisnuson@cri.or.th (J.S.); 4Department of Biochemistry and Molecular Biology, University of Melbourne, Parkville, VIC 3010, Australia; yrustam@student.unimelb.edu.au; 5School of Chemistry, Bio21 Molecular Science and Biotechnology Institute, University of Melbourne, Parkville, VIC 3010, Australia

**Keywords:** glycolipids, sphingolipids, cerebrosides, ceramides, lipidomics, enzymology

## Abstract

Glucosylceramide (GlcCer) is a major membrane lipid and the precursor of gangliosides. GlcCer is mainly degraded by two enzymes, lysosomal acid β-glucosidase (GBA) and nonlysosomal β-glucosidase (GBA2), which may have different isoforms because of alternative splicing. To understand which GBA2 isoforms are active and how they affect glycosphingolipid levels in cells, we expressed nine human GBA2 isoforms in COS-7 cells, confirmed their expression by qRT-PCR and Western blotting, and assayed their activity to hydrolyze 4-methylumbelliferyl-β-D-glucopyranoside (4MUG) in cell extracts. Human GBA2 isoform 1 showed high activity, while the other isoforms had activity similar to the background. Comparison of sphingolipid levels by ultra-high resolution/accurate mass spectrometry (UHRAMS) analysis showed that isoform 1 overexpression increased ceramide and decreased hexosylceramide levels. Comparison of ratios of glucosylceramides to the corresponding ceramides in the extracts indicated that GBA2 isoform 1 has broad specificity for the lipid component of glucosylceramide, suggesting that only one GBA2 isoform 1 is active and affects sphingolipid levels in the cell. Our study provides new insights into how increased breakdown of GlcCer affects cellular lipid metabolic networks.

## 1. Introduction

Sphingolipids help maintain the integrity of membrane structure and organization, and have been implicated in metabolism, cell signaling, neurodevelopment, inflammation, cancer, and several other physiological and pathological processes [1,2,3,4,5,6]. De novo synthesis of ceramide occurs in the endoplasmic reticulum (ER), and ceramide then serves as a precursor for synthesis of hexosylceramides, sphingomyelin, and sphingosine [1,7]. Hexosylceramides, also known as cerebrosides, include glucosylceramide (GlcCer) and galactosylceramide (GalCer), which serve as precursors for the synthesis of more complex glycosphingolipids.

A portion of the ceramide may be converted to GalCer by galactosylceramide synthase in the ER in certain tissues (mainly brain myelin) [1]. The remaining ceramide is transported to the Golgi complex, where one of two enzymes catalyzes the synthesis of two complex sphingolipids, GlcCer and sphingomyelin. Transfer of phosphorylcholine from phosphatidylcholine (PC) to Cer on the lumen side of the Golgi membrane by sphingomyelin synthase (SMS) produces sphingomyelin, while on the cytosolic side of the Golgi membrane, glucosylceramide synthase (GCS adds a glucosyl group from UDP-glucose to ceramide to make glucosylceramide [6]. Furthermore, in the Golgi GlcCer can act as the building block for lactosylceramide and other complex glycosphingolipids, such as gangliosides [6].

GlcCer and its more complex products are ultimately fated to undergo hydrolytic degradation by lysosomal glucosylceramidease (acid β-glucosidase, GBA, E.C. 3.2.1.45) and glucosylceramidase 2 (nonlysosomal β-glucosidase, GBA2, E.C. 3.2.1.45) to release ceramide and glucose [7,8,9,10]. GBA dysfunction causes the accumulation of GlcCer, and results from *GBA* mutations in Gaucher disease or may be triggered as a secondary defect of the accumulation of other lipids, such as cholesterol and GM1 and GM2 gangliosides [6,7,11,12,13]. Deficiency of GBA has been implicated in the etiology of Parkinson’s disease [14]. In contrast, humans carrying mutations in the *GBA2* gene are affected with cerebellar ataxia with spasticity or spastic paraplegia (Spastic Gait locus #46, SPG46) [15,16].

GBA2 was first described as a nonlysosomal glucosylceramidase activity that was tightly associated with membranes and exhibited a higher pH optimum and different inhibition profile than GBA [17]. It was later cloned as human bile acid β-glucosidase [18], a 100 kDa protein that was unrelated to GBA and originally not found to hydrolyze glucosylceramide [19]. However, investigation of knockout mice showed that it affected glucosylceramide levels rather than bile acid glucoside levels [20,21]. It was also observed that overexpression of GBA2 in COS-7 cells resulted in increased conversion of fluorescent GlcCer to ceramide and sphingomyelin [22], as was previously observed for its native activity [17]. In addition to hydrolysis of GlcCer and bile acid glucosides, GBA2 has been reported to transglycosylate cholesterol using GlcCer and GalCer as donors [23,24]. In contrast to the lysosomal GBA, GBA2 is a nonintegral membrane protein associated with the cytoplasmic side of the ER and/or Golgi [25]. The metabolic interaction of GBA2 with GBA was demonstrated through the improved aspect of GBA knockout mice and Neimann-Pick model mice, in which GBA deficiency is a secondary effect, when GBA2 was also knocked out [26,27].

Alternative splicing generates different transcripts from the same gene and affects the expression levels, stability, half-life, and localization of the RNA messengers [28,29]. It has the potential to generate several protein isoforms with different biological properties, protein–protein interactions, subcellular localization, signaling pathway, or catalytic ability. Although several putative splice isoforms exist for human GBA2 in the National Center for Biotechnology Information (NCBI) database Gene ID 57,704 entry, only the predominant isoform 1 (NP_001317589.1) has been characterized so far.

In order to understand the biological significance of GBA2 isoforms on sphingolipid metabolism, we have studied GBA2 isoforms by transient overexpression of isoforms predicted by RNA sequencing in mammalian cells, followed by activity assay with artificial substrate (4MUG) and assessment of lipid changes in the cells, including that of the natural substrate (GlcCer) using semi-quantitative ultra-high resolution/accurate mass spectrometry. These findings provide further evidence of the effect of GBA2 on sphingolipid metabolism. 

## 2. Results

### 2.1. Sequence and Structure Analysis of GBA2 Isoform

Thirteen GBA2 isoforms are listed in the National Center for Biotechnology Information (NCBI) Gene database Gene ID 57,704 entry. Experimentally determined cDNA sequences for only two of these, isoform 1 and isoform X1, are found in the database, while the rest are supported by high throughput mRNA sequencing data (RNASeq). We have evaluated the functionality of the nine of these GBA2 isoforms that cover most of the gene and do not contain other start codons before their putative start codons by analyzing their effects on the putative structure, and by expression of the corresponding cDNA in COS-7 cells. 

GBA2 isoform protein sequences are aligned in Figure 1. Isoform 1 is the well-characterized standard form of GBA2 (NP_001317589.1), while isoform X1, for which a cDNA has been isolated from substantia nigra (NCBI: AK295967.1), differs only by the insertion of six amino acid residues in the N-terminal domain (NP_001317589.1:p.Q189_F190insPICPLK). Isoforms X2 and X4 share this same insertion. Isoforms 2, X4, X6, and X8 have an alternative C-terminus (p.G842_S877delinsX27), which is shorter than that found in isoform 1. Isoforms X2, X3, and X6 are missing 22 amino acid residues (p.V634_L655del) that contribute to two helices and a loop around the active site in the model of the human GBA2 structure (Figure 2, Appendix A) [30]. Isoforms X7 and X8 are missing 79 amino acid residues (p.F190_Q268del), which comprise 5 β-strands in the N-terminal domain. In order to closely inspect parts that are missing or added in various isoforms, the other isoforms were aligned to isoform 1, as shown in Appendix A. To clarify this further, the missing or added structures are shown next to their specific position in the alternative isoform models in Appendix A. Since all of these isoform differences could potentially lead to activity differences, we tested their activities in COS-7 cells.

### 2.2. GBA2 Isoform Expression and Activity of Human GBA2 Isoforms in Transfected COS-7 Cells

We chose COS-7 (Green monkey kidney) cells to evaluate the effects of the alternative splicing-generated sequence variations on GBA2 activity its effect on sphingolipids, because COS-7 cells have been shown to be an effective system for expression of GBA2 [22,32]. They were also reported to have undetectable levels of galactosylceramide [33], so that hexosylceramide levels should primarily reflect glucosylceramide. Expression analysis by qRT-PCR confirmed that the mRNA levels of all human GBA2 isoforms were not significantly different in the same incubation time, while they were significantly decreased at 72 h compared to 48 h (Figure 3A). We then confirmed the expression of each human GBA2 isoform by Western blotting using anti-GBA2 and anti-FLAG-tag antibodies (Figure 3B,C), which showed that all human GBA2 isoforms were expressed at the protein level at both 48 h and 72 h. In order to identify which human GBA2 isoforms were active, activity against the fluorescent substrate 4-methylumbelliferyl β-D-glucoside (4MUG) was measured in lysates from transfected COS-7 cells (at 48 and 72-h post-transfection). CBE was added as a GBA inhibitor to one set of assays, although it also exhibits weak inhibition of GBA2 [34]. After 48 h of transfection, we found that human GBA2 activity of cells transformed with isoform 1 was significantly higher than cells transfected with empty vector as a negative control, while cells with other isoforms were similar to the negative control. Seventy-two hours after transfection, the human GBA2 activity of cells transfected with isoform 1 was significantly lower than after 48 h (Figure 3D). Furthermore, GBA2 activity decreased about 25% upon addition of CBE, confirming that most of the activity resulted from GBA2 rather than endogenous GBA activity. Thus, all human GBA2 isoforms were expressed but only isoform 1 clearly hydrolyzed the MUG substrate. These results suggest that none of the deletions or insertions shown in the protein sequence alignment in Figure 1 and structural model (Figure 2, Appendix A) can be accepted and still form an active 4MUG hydrolase.

### 2.3. Analysis of Sphingolipid Levels in COS-7 Cells Overexpressing GBA2 Isoforms

The above results confirmed the GBA2 isoform 1 expressed in COS-7 cells can hydrolyze the synthetic 4MUG substrate, while other isoforms showed little or no activity with this substrate. However, this did not indicate whether human GBA2 isoform 1 or the other isoforms can act on natural glucosylceramide substrates in the cells. Positive ionization mode ultra-high resolution accurate mass spectrometry (UHRAMS) spectra were acquired from transfected COS-7 cells. The mass spectra of the lipid extracts of COS-7 cells expressing GBA2 isoform 1 and cells with empty vector control are compared in Appendix A, which shows that they are similar. However, when individual mass peaks corresponding to three species of sphingolipid, including hexosylceramide, ceramide and sphingomyelin with 34:1, 40:1, 42:1, and 42:2 ceramide carbon lengths and desaturation levels were compared, significant differences are observed (Appendix A). The levels of total sphingolipid species identified at the 48-h and 72-h time points, including total sphingolipid (total of all Cer, Hex1Cer, Hex2Cer, ganglioside (GM) and SM species), total ceramide, total hexosylceramide, and total sphingomyelin, are shown in Appendix A. It should be noted that HexCer levels are relative rather than absolute levels, since they were normalized to the ceramide internal standard, because of the lack of HexCer internal standards in this study, but these can nonetheless indicate the relative differences between the samples. Total ceramide was not significantly increased by GBA2 isoform 1 overexpression, while total hexosylceramide decreased, albeit not by a significant amount, and total sphingolipid and sphingomyelin underwent an insignificant increase in cells expressing GBA2 isoform 1 compared to the control. However, this analysis included sphingolipids that were detected in some samples and not others. The heat map of the sphingolipid concentration Z-scores at 48-h post-transfection in Figure 4A shows the ceramide (Cer), hexosyl ceramide (Hex1Cer) and sphingomyelin lipids that were identified in all conditions. The identified ceramides (Cer) had total numbers of carbons of C32–42, including the sphingoid base and long-chain fatty acid, while identified hexosyl ceramides (Hex1Cer) had lipid components of C38–42, and identified sphingomyelins (SM) included species with C32–44. Interestingly, the pattern of extracts of cells expressing GBA2 isoform 1 had a clearly unique pattern and it was identified as the outgroup by clustering of the cell extracts based on their sphingolipid compositions, which is what would be expected if it is the only isoform with significant activity on the sphingolipids. 

In the extract from cells transformed with isoform 1, the mono-hexosylceramide species Hex1Cer (34:1), Hex1Cer (40:1), Hex1Cer (42:1), and Hex1Cer (42:2) showed low abundances (low Z-scores), while ceramides Cer 34:1, 40:1, 42:1, and 42:2 were found at high levels relative to the other conditions. The heat map also indicated that SM species 34:2, 36:2, 40:2, 42:1, 42:2, 42:3, 44:2, and 44:3 were higher in isoform 1-expressing cell extracts than in those of other cells. However, other SM species, such as d32:1, d36:1, and d38:1, were at similar or lower levels in isoform 1 expression cell extracts compared to those of other cells. This suggests that GBA2 isoform 1 hydrolyzes GlcCer to release Cer and glucose, resulting in lower hexosylceramide and higher free ceramide levels, while other isoforms did not significantly affect these levels, as emphasized by combining all species of each lipid class in the bar graphs in Figure 4B,C. In contrast, the levels of Hex1Cer (d38:2), and the di-hexosyl (i.e., lactosyl) ceramide species Hex2Cer (34:1), Hex2Cer (42:1), and Hex2Cer (42:2) were similar to or slightly higher than those in the control upon overexpression of GBA2 isoform 1. This suggests that depletion of the glucosylceramide did not have a significant effect on dihexosyl ceramide levels. The differences between SM levels in control cells and cells expressing isoform 1 were not significant, as shown in Figure 4D. In contrast to isoform 1, the other isoforms did not induce significant changes in cellular sphingolipid levels. These results confirmed that GBA2 isoform 1 is the only active isoform and indicate that Hex1Cer isoforms 34:1, 40:1, 42:1, and 42:2 all appear to be human GBA2 substrates. 

In contrast, the heat map of the sphingolipid concentration Z-scores at 72-h post-transfection was not significantly different in all conditions, as shown in Appendix A. This result suggests that other factors had more influence on the sphingolipid levels than expression of GBA2 as its activity decreased.

### 2.4. Analysis of Sphingolipid Ratios Related to the Direction of Sphingolipid Metabolic Flow

To generate a more sensitive parameter for the movement of ceramides from glucosylceramides to other species upon overexpression of GBA2, the ratios of Cer to Hex1Cer and of SM to Hex1Cer were calculated. Hex1Cer and Cer with the same ceramide masses that were detected included 34:1, 40:1, 42:1, and 42:2 species, while of these SM species were detected only for 40:1 42:1, and 42:2. The expression of human GBA2 isoform 1 in COS-7 cells resulted in an increased ratio for the Cer to Hex1Cer (34:1, 40:1, 42:1, and 42:2) compared to the ratio for the empty vector control (Figure 5A). However, expression of the other isoforms did not significantly change these ratios at 48 h, as seen in Figure 5A. The ratio of SM to Hex1Cer (40:1, 40:2, and 42:2) was also significantly higher in GBA2 isoform 1 transfected COS-7 cells, compared to other isoforms and to control (Figure 5B). This evidence suggests that GBA2 activity contributes to the conversion of sphingolipid from GlcCer to Cer, and that some of the released Cer may be subsequently converted to SM, as previously noted with exogenously added fluorescent GlcCer [17,22]. 

Since a GBA2 isoform could act on one or a subset of the glucosylceramide species, the change in intracellular sphingolipid ratio was analyzed for each ceramide/hexosylceramide pair, including Hex1Cer to Cer (34:1) in Figure 5C, Hex1Cer to Cer (40:1) in Figure 5D, Hex1Cer to Cer (42:1) in Figure 5E, and Hex1Cer to Cer (42:2) in Figure 5F. The ratios of Cer to Hex1Cer were increased for cells transfected with GBA2 isoform 1, but not with the other isoforms. At 72 h, the ratio of Cer to Hex1Cer and SM to Hex1Cer were higher, but not significantly different from control because of high variation, as shown in Appendix A. These results suggest that each pair represents a GBA2 isoform 1 substrate and product, while the other isoforms show no obvious activity toward any of them.

### 2.5. Analysis of Total Lipid Composition

The total lipid distribution among the three lipid classes, including sphingolipids, glycerophospholipids, and glycerolipids did not change significantly in cells overexpressing human GBA2 isoform 1 compared to control, as shown in Figure 6 (48-h post-transfection) and Appendix A (72-h post-transfection). Consideration of the amounts of individual species of each lipid class in Figure 6B–D shows that sphingolipids and glycerolipids appeared to increase slightly, while the average amounts of phosphoglycerolipids did not change significantly. These results suggest that overall lipid homeostasis was not disturbed by human GBA2 overexpression.

### 2.6. Analysis of Glycerophospholipids/Glycerolipids Involved in Sphingolipid Metabolism

In the previous analysis, Cer was increased in response to overexpression of GBA2 glucosylceramidase, and this appeared to cause an increase in some SM levels as well. Since SM synthase transfers a phosphocholine group from phosphotidylcholine (PC) to Cer to produce SM and diacylglycerol (DAG) [4,5,35,36], we analyzed the ratio of total DAG to PC, at 48-h post-transfection in Figure 7A. DAG and PC species with fatty acyl components of 34:1, 40:2, 40:3, 40:5, and 40:6 (number of carbons: double bonds) were detected in both control and GBA2-overexpressing cell extracts. The ratio of DAG to PC was increased for the total of all of these species and for each species independently in cells overexpressing GBA2 isoform 1 compared to control. However, this increase was only significant for DAG to PC with the fatty acyl component 34:1, while no significant differences were observed at 72 h (Appendix A). In comparison, no significant differences were observed in the ratios for DAG to PE and PI (Appendix A).

## 3. Discussion

GBA2 deficiency is responsible for a heterologous group of ataxias, including hereditary spastic paraplegia (HSP), autosomal recessive cerebellar ataxia (ARCA) with spasticity and Marinesco-Sjogren-like syndrome [15,16,37]. The patients with *GBA2* mutations have progressive diseases that can present from early childhood to early adulthood and have a wide range of neurological and non-neurological symptoms, including cataracts, hypotonia, brisk tendon reflexes, gait ataxia, spasticity, and mental impairment, with thinning of the corpus callosum and atrophy of the cerebellum. The molecular basis of how GBA2 deficiency leads to these syndromes and the basis for their heterogeneity are not well understood. 

We considered the possibility that some splice isoforms might show differential tissue expression and certain mutations might affect some of these more than others. Although previous papers only considered GBA2 isoform 1 [8,15,20,22,32,34,35], twelve other isoforms are predicted from RNA sequencing (RNA Seq) data in the NCBI database, with one resulting in a seemingly mild change in the noncatalytic domain and others that might be accommodated by slight structural adjustments in the catalytic domain (Figure 2, Appendix A). However, when we expressed the nine human GBA2 isoforms with the most complete sequences, only isoform 1 had activity toward MUG and caused a significant change in cellular sphingolipids. Surprisingly, isoform X1, which has an insertion of six amino acid between residues 189 and 190 in a β-strand of the N-terminal non-catalytic domain compared to isoform 1, showed no significant activity, indicating that the structural integrity of the N-terminal domain is also critical to the activity. 

It is unclear why the small insertion of six amino acids in the N-terminal domain of isoform X1 disrupts the activity of GBA2 or whether this isoform has a function. Although a database entry exists for an experimental cDNA for this isoform, as mentioned in the Results, that work has not been published and the prevalence of that isoform is unknown. It is notable that the N- and C-terminal domains in the GH116 family are strongly linked with a long α-helix shared between the two domains and the loop from the N-terminus helps form the entrance to the *Tx*GH116 β-glucosidase structure and in the human GBA2 homology model [30]. Truncations of *Tx*GH116 beyond the N-terminal hydrophobic segment (possible signal sequence) did not allow the production of a functional C-terminal catalytic domain. On the other hand, overexpression of the protein with bulky C-terminal tags, such as green fluorescent protein (GFP), gives localization of the protein to the plasma membrane [22,38], suggesting that the C-terminus might be important for ER/Golgi localization. Severe truncations seen in some human mutations result in localization of the protein fragments to the mitochondria, although nonsense mediated decay of the mRNA is likely to limit the effects of these aberrant proteins on the patients [38]. The splicing isoforms are natural rather than mutation induced, but isoforms 2, X4, X6, and X8 do have an alternative C-terminus that might affect localization. Mutation of GBA2 was also shown to effect the cytoskeleton, which could reflect noncatalytic roles of GBA2, in addition to the catalytic role of GBA2 isoform 1 in mediating sphingolipid levels [39]. A variant in GBA has been shown to cause alternative splicing leading to Gaucher disease [40], so the possibility of GBA2 mutations leading to over production of inactive isoforms should be considered. Although nonpathological variants of GBA2 have rarely been described [41], recently it was reported that GBA2 variants are predictive for response to chemotherapy in non-small-cell lung cancer [42]. Further studies should be considered to see whether these variants affect activity or alternative splicing.

From the NCBI Gene page collected RNA Seq data, GBA2 (Gene ID: 57704) is most highly expressed in small intestine, duodenum, kidney thyroid, colon, and brain. Although there is a database entry for a cDNA from substantia nigra mRNA for isoform X1, that does not really indicate the expression level of this isoform in that tissue. GBA2 has been noted to be under expressed in neurons in ataxia datasets, although alternative splicing was not considered [43]. In the future, it would be interesting to analyze the expression of specific isoforms in human tissues. The mRNA for the isoforms that we expressed do not have stop codons positioned for nonsense-mediated decay, but this does not rule out that they might be processed to small regulatory RNA or otherwise serve to modify GBA2 isoform 1 mRNA levels in certain human cells where they are expressed. 

Although previous studies have expressed GBA2 in cells, including studying the effects of mutations [36], most looked at the activity on the synthetic substrate MUG. In other cases, the effect of deficiency of GBA2 on sphingolipid levels was explored in animals [20,26,27]. For instance, glucosylceramides with various lipid components were found to build up in testis and dermal fibroblasts of homozygous GBA2 knockout mice [26]. Clearly, loss of GBA2 activity affects the glucosylceramide levels, which could also affect levels of more complex glycosphingolipids, although no significant decrease in ceramide was observed. Boot et al. [22] showed that GBA2 overexpressed in COS-7 cells could break down exogenously added synthetic fluorescent glucosylceramide substrate, and some of the released ceramide could be metabolized to sphingomyelin, which recapitulated the observed effect of native GBA2 in human cells on the same substrate [17]. Consistent with those results, overexpression of GBA2 isoform 1 in COS-7 cells in this work resulted in decreased levels of Hex1Cer species, corresponding to the hydrolysis of glucosylceramides. In contrast to that previous work in mice, we detected a clear increase in ceramides, while no significant decrease in ceramides was seen in GBA2 deficiency in mouse testes or fibroblasts [26]. Similar increases in GlcCer were seen in the brains of Niemann-Pick Type C model mice, when GBA2 was knocked out or inhibited [27]. Glucosylceramides were also significantly higher in lymphoblastoid cells from a patient with a homozygous GBA2 mutation compared to control lymphoblastoid cells [44]. So, in general, our observations in overexpression of GBA2 on sphingolipids are opposite to those seen in animal and human cells with GBA2 deficiency, as expected, except that we also detected a significant change in ceramide levels.

Upon comparing the general levels of hexosylceramides in Figure 4, the decrease seen upon overexpression of GBA2 isoform 1 was not significant, partly because of the presence of species that were detected in some samples and not others. By comparing only species of lipids found as both ceramides and hexosylceramides in all samples in Figure 4B,C, we were able to see a significant change. The Figure 4A heat map showed that Hex1Cer (42:1), (42:2), (34:1), and (40:1) increased significantly compared to control, while Hex1Cer (38:2) was similar in cells with overexpressed GBA2 isoform 1 and control. This suggests that Hex1Cer (d38:2) may have a higher fraction of hexosylceramides that could not be hydrolyzed, while the other four Hex1Cer masses represented higher fractions of glucosylceramides that could be hydrolyzed by GBA2. Despite the obvious SM increase seen upon overexpression of GBA2 isoform 1 in the Figure 4A heat map, the change was not to a significant level. To develop a more sensitive parameter, we evaluated the relative ratio of ceramide to hexosylceramide for lipid species found in both classes in Figure 5, and found that the difference upon GBA2 isoform 1 expression was much more highly significant. The ratio of sphingomyelins to hexosylceramides of the same species showed a similar level of significance (Figure 5A), suggesting that some hexosylceramide hydrolyzed by the overexpressed GBA2 is likely to be converted to sphingomyelin, consistent with previous observation of conversion of fluorescent glucosyl ceramide conversion to ceramide then SM [17,22]. 

Since sphingomyelin synthase transfers phosphocholine from PC to ceramide to make SM and release DAG, we also investigated the ratios of DAG to PC with the same fatty acid masses. As seen in Figure 7, the ratios all increased for those species detected in both PC and DAG, although only in the case of the 34:1 species (likely corresponding to one oleic acid (18:1) and one palmitic acid (16:0)) was the increase statistically significant. The overall lipid proportions were not disrupted (Figure 6), and no significant differences in levels of total glycerophospholipids, phosphatidylcholine, phosphatidyl ethanolamine, phosphatidlylinositol and glycerolipids were observed compared to other isoforms and empty vector, as shown in Appendix A. The results suggest that overall lipid homeostasis was not generally disrupted by the GBA2 overexpression, but the levels of certain DAG species may have increased, along with Cer and SM, since PC levels are high and unlikely to be significantly affected by their use for SM synthesis. Given the role of DAG in protein kinase C activation and signaling [27], this could be another aspect of GBA2 deficiency or excess GBA2 activity. 

A case of overexpression of GBA2 has been reported in the spinal cord of a superoxide dismutase mutant mouse model for amyotrophic lateral sclerosis (ALS), a fatal disease resulting in loss of motor neuron function [45]. Treatment with the GBA2 inhibitor ambroxol hydrochloride delayed loss of muscle strength and death in this model, and could also promote neuromuscular junction formation in tissue culture. GBA2 activity is increased by high substrate concentrations when lysosomal GBA is deficient in Gaucher and Niemann–Pick models, some symptoms of which are decreased when GBA2 is knocked out or inhibited [26,27]. These symptoms were suggested to be caused by the release of sphingosine in the cytoplasm, but our data suggest that changes in ceramide, glucosylceramide, and DAG levels should also be considered. In contrast to these cases where knocking down GBA2 activity is desired, mutations that knock out GBA function lead to hereditary spastic paraplegia, autosomal recessive ataxia with spasticity and Marinesco-Sjogren-Like Syndrome [15,16,37,39]. Therefore, it is clear that the amount of GBA2 activity needs to be carefully controlled to allow proper maintenance of the central nervous system, as well as peripheral tissues.

## 4. Materials and Methods

### 4.1. Reagents

Dulbecco’s modified eagle medium, penicillin/streptomycin (Pen Strep), trypsin/EDTA, and fetal bovine serum were obtained from Thermo Fisher Scientific (Waltham, MA, USA). The coding sequences for nine isoforms of human GBA2: major transcript isoform 1 (NM_020944.3), isoform 2 (NM_001330660.1), isoform X1 (XM_006716809.3), isoform X2 (XM_005251526.4), isoform X3 (XM_017014937.1), isoform X4 (XM_017014938.1), isoform X6 (XM_017014940.1), isoform, X7 (XM_017014941.1), and isoformX8 (XM_017014942.2) were synthesized and inserted into the pcDNA3.1+/c-(k)-dyk expression vector for mammalian cells by GenScript Corporation (Piscataway, NJ, USA). The human GBA2 peptide CRRNVIPHDIGDPDD was synthesized and an anti-peptide antibody to it was also generated at Genscript Corp. Anti-β-actin and anti-Flag-tag antibodies were from Cell Signaling Technology (Danvers, MA, USA). Conduritol-β-epoxide and 4-methylumbelliferyl-β-D-glucuronide were from Sigma-Aldrich (St. Louis, MO, USA). Deuterated internal standard lipids phosphatidylcholine (PC 15:0/18:1-d7), phosphatidylethanolamine (PE 15:0/18:1-d7), phosphatidylserine (PS 15:0/18:1-d7), phosphatidylglycerol (PG 15:0/18:1-d7), phosphatidylinositol (PI 15:0/18:1-d7), phosphatidic acid (PA 15:0/18:1-d7), lysophosphatidylcholine (LPC 18:1-d7) lysophosphatidylethanolamine (LPE 18:1-d7), cholesterol ester (18:1-d7), monoacylglycerol (MG 18:1-d7), diacylglycerol (DG 15:0/18:1-d7), triacylglycerol (TG 15:0/18:1-d7/15:0), SM (d18:1/18:1-d9), and Cer (d18:1-d7/15:0) were from Avanti Polar Lipids (Alabaster, AL, USA). Ammonium formate was from Alfa Aesar (Ward Hill, MA, USA). Propanol, methanol, and water were from J.T. Baker (Phillipsburg, NJ, USA) and chloroform was from EMD chemicals (Billerica, MA, USA). All solvents used were of high-performance liquid chromatography grade, and all lipid extraction and storage solvents contained 0.01% butylated hydroxytoluene (BHT) from Sigma-Aldrich.

### 4.2. Cell Culture and Transfection

COS-7 cells (African green monkey kidney) were seeded into 75 cm^2^ cell culture flasks with Dulbecco’s Modified Eagle Medium (DMEM) containing 10% Pen-Strep and 10% fetal bovine serum. Cell lines were grown in a 5% CO_2_ incubator at 37 °C. One million COS-7 cells were seeded into a 6 cm^2^ plate with DMEM medium, and then the COS-7 cells were cultured to 80–90% confluency overnight. The medium was removed and the plate washed once with sterile 1X phosphate-buffered saline (PBS), then 2 mL of Opti-MEM (reduced serum medium, Thermo Fischer Scientific) was added into the plate. COS-7 cells were transfected with the cDNA-encoding human GBA2 isoforms in the pcDNA3.1+/C-(k)-dyk mammalian expression vector with lipofectamine 2000 reagent (Thermo Fischer Scientific, Waltham, MA, USA), according to manufacturer’s instructions. After 6 h, the Opti-MEM was removed, and replaced by DMEM complete medium with 1% Pen Strep. COS-7 cells were incubated for 48 h or 72 h, then the cells were washed with PBS, scraped in PBS, and stored at −80 °C until use.

### 4.3. RNA Extraction and Quantitative RT-PCR

COS-7 cells were collected at 48 h and 72 h after transfection. RNA was extracted in Trizol reagent (Thermo Fischer Scientific, Waltham, MA, USA), according to manufacturer’s instructions. The first stand cDNA was generated by SuperScript™ III Reverse Transcriptase cDNA synthesis Kit (Thermo Fischer Scientific), and the synthesized cDNA synthesis reaction was stored at −20 °C until use. RT-PCR was performed using SYBR green/Rox qPCR master mix (Thermo Fischer Scientific) on the LightCycler^®^ 480 II Instrument (Roche Molecular Systems, Inc., Pleasanton, CA, USA). The primers for qPCR were GBA2-Forward: 5′-CCACTACAGGCGGTATACAA-3′ and GBA2-reverse: 5′-GATCTGTCATCCAATACCGG-3′, and β-actin-Forward: 5′-GATCAGCAAGCAGGAGT ATGACG-3′ and β-actin-reverse: 5′-AAGGGTGTAACGCAACTAAGTCATAG-3′.

### 4.4. Protein Collection and Western Blotting Analysis

The medium was removed and cells washed two times with ice-cold 1X PBS, then the COS-7 cells were collected in 1 mL of ice-cold 1X PBS by scraping, and the cell suspension was transferred to a 1.5-mL tube on ice. Then, the cell suspension in 1X PBS was sonicated on ice. The extracted cells were diluted 1:20 in 1X PBS and the protein concentration was measured with a Pierce^TM^ BCA Protein Assay kit from Thermo Fischer Scientific (#23225). Proteins were separated by SDS-PAGE using the Criterion system (BioRad, Hercules, CA, USA) and transferred to nitrocellulose membrane by wet Western blotting transfer in 50 mM Tris-base, 40 mM glycine, and 20% methanol. Blots were blocked by 5% skimmed milk in 0.05% PBST for 1 h, and washed with 0.05% PBST, then incubated with anti-GBA2 antibody (1:100), anti-FLAG antibody (1:1000), or anti-β-actin antibody (1:2000) as primary antibody overnight. After washing three times with PBST, goat anti-rabbit/HRP (Genscript) and rabbit anti-mouse/HRP (DAKO)-conjugated secondary were incubated with the blots to detect GBA2 (rabbit polyclonal antibodies) and Flag-tag primarily (rabbit monoclonal antibodies) and β-actin (mouse monoclonal antibodies) antibodies, respectively. After washing three times with PBST, the blots were developed with Luminata Forte Western HRP Substrate, according to the manufacturer’s instructions (Merck, Kenilworth, NJ, USA).

### 4.5. Measurement of GBA2 Enzyme Activity on MUG

GBA2 enzyme activity was assayed as described elsewhere [29,36,38]. Samples were pre-incubated with or without 10 μM CBE, followed by incubated with the 4-methylumbelliferyl-β-D-glucoside (4MUG) substrate, 3.5 mM final concentration (Sigma-Aldrich, St. Louis, MI, USA) at pH 5.8 and 37 °C for 30 min.The reactions were terminated by adding 200 µL 1 M of glycine, pH 10.6, then the fluorescent signal was measured in a fluorescence microplate reader with excitation at 355 nm and emission at 460 nm.

### 4.6. Lipid Extraction and Lipid Measurement

Cell pellets and extraction blank were freeze dried overnight and stored in −80 °C until use. The samples were subjected to monophasic methanol/chloroform/water lipid extraction, as previously described [25,46]. The supernatants were transferred to 2.0 mL glass vials, and stored at −80 °C until further use. Ten microliters of lipid extracts were evaporated, then washed with 10 mM NH_4_HCO_3_, followed by reconstitution in 40 µL of isopropanol: methanol: chloroform (4:2:1, *v*:*v*:*v*), containing 20 mM ammonium formate. The solutions were then placed into the wells of an Eppendorf twin-tec 96-well PCR plate (Eppendorf, Hamburg, Germany), and the plate was sealed with sealing tape. Samples were then aspirated via direct infusion nanoESI into an ultra-high resolution/accurate mass Thermo Scientific model Orbitrap Fusion^TM^ Lumos^TM^ Tribrid^TM^ mass spectrometer with an Advion Triversa Nanomate nESI source (Advion, Ithaca, NY, USA), operating with a spray voltage of 1.2 kV in positive mode and 1.4 kV in negative mode, and a gas pressure of 0.3 psi, as described [47,48,49]. For the mass spectrometer, the ion transfer capillary temperature was set to 150 °C, the radio frequency (RF)-value to 10%, and the AGC target to 2 × 10^5^. Spectra were acquired at a mass resolving power at 500,000 (at 200 m/z). Peaks corresponding to the target analytes and internal standards (ISs) were identified by automated peak finding and then assigned at the “sum composition” level of annotation using a developmental version of Lipid Search 5.0α software (Mitsui Knowledge Industry (MKI), Tokyo, Japan, and Thermo Fisher Scientific) by searching against an accurate mass-based, user-defined database. The search parameters were parent (noise) Threshold:150:Parent (mass) tolerance: 1.5 ppm, Correlation threshold (%): 0.3, Isotope threshold (%): 0.1, Max isotope number: 1 (i.e., including the M+1 peak). Peak detection was set to profile and merge mode to average. The internal standards were used to calibrate the mass spectra prior to database searching. Semi-quantitative analysis of identified endogenous lipids was performed by comparison of their peak areas to the peak areas of the relevant internal standards. The SM internal standard was used for sphingomyelin species and the Cer internal standard was used for ceramide and hexosylceramide species, and by further normalizing to total protein (µg). Note that at the level of annotation achieved using this acquisition and analysis method, glucosylceramide (GlcCer) and galactosylceramide (GalCer) lipids may not be not differentiated from each other, so are collectively assigned here as hexosylceramides. However, since COS-7 cells were reported to contain undetectable levels of GalCer [33], the HexCer levels likely correspond to GlcCer.

### 4.7. Sequence and Structure Analysis

Nine GBA2 isoforms listed in the National Center for Biotechnology Information (NCBI) Gene database entry Locus 57704, including isoform 1, isoform 2, isoformX1, isoformX2, isoformX3, isoformX4, isoformX6, isoformX7, and isoformX8, were aligned in MEGA10. Homology modeling was done in the SWISS-MODEL server (https://swissmodel.expasy.org) [31] with the *Tx*GH116 β-glucosidase structure as template (30), and models were visualized in PyMOL (Schrödinger LLC, Portland, OR, USA).

### 4.8. Data and Statistical Analysis

Results are expressed as the mean ± SD of three independent biological replicates. The output data was visualized via GraphPad Prism 8.0.2 (GraphPad Software, San Diego, CA) and R Core Team (2019; R: A language and environment for statistical computing. R Foundation for Statistical Computing, Vienna, Austria. URL https://www.R-project.org/) order script, setwd(‘’) to create the file directory, Lipid <- read.csv(‘file nmae.CSV’, row.name = 1) to import the. CSV file to R, library(“pheatmap”) to download the library algorithm from the database, pheatmap(Lipid, cutree_rows = 4) to generate the heat map with clustering analysis to group similarities in the heat map. All values were expressed as Z-score (the position of a raw score in terms of its distance from the mean, z = (x − μ)/σ, where x is the mean value for samples of the same isoform, μ is the mean of the values of all isoforms and σ is the standard deviation between the means of all isoforms). For statistical analysis, the distributions of sample values were evaluated for deviance from a normal distribution by the Shapiro-Wilk test [50]. The differences in the means of RNA expression, activity determination, abundance of lipid classes and individual lipid species were compared between empty vector control and cells expressing GBA2 isoforms by a two-tailed unpaired *t*-test via GraphPad Prism 8.0.2 software. In the case of expression and activity data, a two-way Analysis of variance (ANOVA) with Tukey’s multiple comparison was made between 48 and 72 h. The mean differences were considered significant at *p* < 0.05 [47].

## 5. Conclusions

Our work has demonstrated that among the possible isoforms predicted from RNA sequencing in human tissues, only GBA2 is likely to affect the cellular lipid levels directly, although we cannot rule out noncatalytic and regulatory roles for other isoforms or their RNA molecules. GlcCer and Cer levels are affected most clearly by GBA2 overexpression, but subtle effects on SM and DAG/PC levels were also seen. Given the effects of these lipids on membrane properties and signaling, GBA2 expression levels may have a significant impact on the cell. The negative effects of loss of GBA2 function in HSP, ARCA, and Marinesco-Sjögren-like syndrome, but positive effects of GBA2 inhibition in Gaucher, Niemann-Pick and ALS models, suggests that GBA2 is part of an important fine-tuning mechanism in lipid metabolism that is particularly critical in neuronal cells. Further work is needed to elucidate the full cellular effects of GBA2 and their pathogenic and therapeutic implications.

## Figures and Tables

**Figure 1 metabolites-10-00488-f001:**
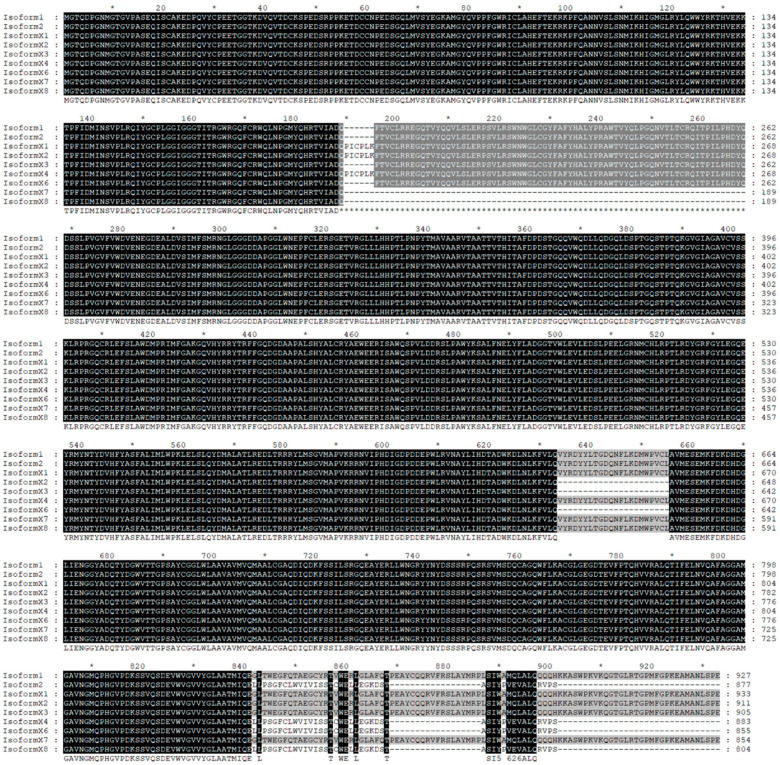
Protein sequence alignment of human GBA2 isoforms.

**Figure 2 metabolites-10-00488-f002:**
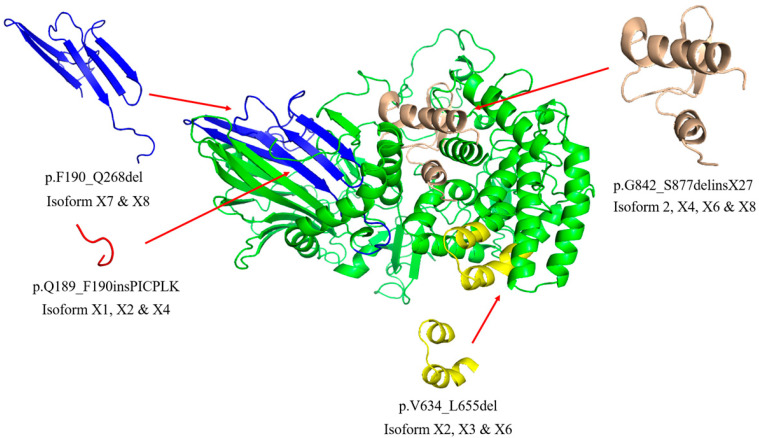
Structural model of human GBA2 highlighting the structures changed by deletions and insertions in the splicing isoforms relative to isoform 1. All structural variations are labeled relative to the human GBA2 isoform 1 reference sequence NCBI: NP_001317589.1. The homology models were generated based on the structure of *Tx*GH116 (PDB: 5BVU) [30] with SWISS-MODEL [31]. The cartoon model of GBA2 isoform 1 (NCBI: NP_001317589.1) is shown in green with deletions shown in various colors with the deleted structures shown in the corresponding color to the side with the list of isoforms containing this variation. The insertion p.Q189_F190insPICPLK predicted structure (red) is positioned based on the isoform X1 model. Individual homology models are shown in Appendix A.

**Figure 3 metabolites-10-00488-f003:**
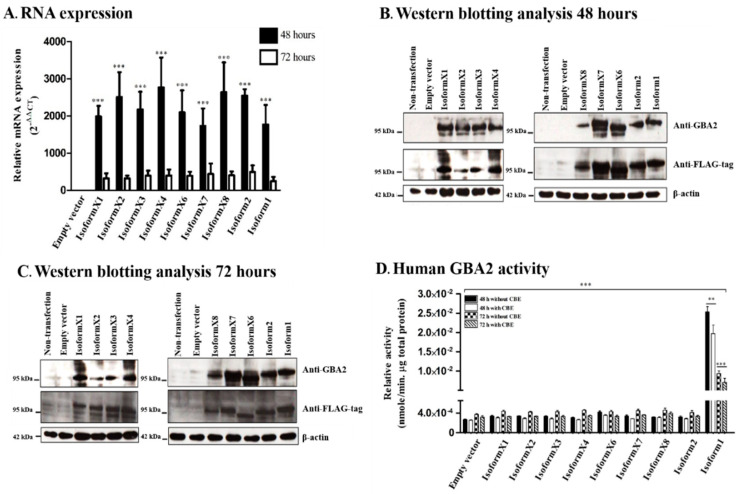
Expression of human GBA2 RNA, protein and activity in COS-7 cells transfected with human GBA2 isoforms. (**A**) RNA expression levels of hGBA2 compared between 48 and 72 h conditions as well as to empty vector control at 48 h; (**B**,**C**) Western blot analysis with different incubation times, 48 and 72 h, using anti-GBA2 and anti-FLAG-tag antibody; (**D**) activity determination of hGBA2 compared between 48 and 72 h as well as between with and without inclusion of CBE. All experiments were done with three independent biological replicates, and means and standard deviations are shown in A and D with * indicating differences with *p* < 0.05, ** indicating *p* < 0.01, *** indicating *p* < 0.001 in a two-tailed unpaired t-test and ANOVA with Tukey’s multiple comparison for different time points. Shapiro–Wilk analysis showed no significant deviance from a normal distribution. For Western blots (**B**,**C**), the most clear example from the three similar replicates of independent biological samples is shown.

**Figure 4 metabolites-10-00488-f004:**
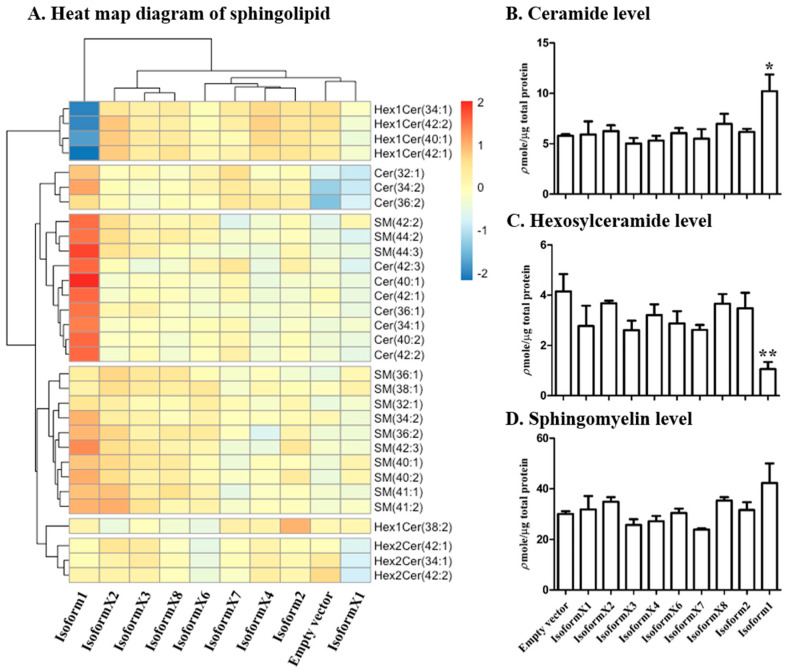
Relative sphingolipid levels in cells expressing respective human GBA2 isoforms. (**A**) The heat map illustrates Z-score differences from mean of sphingolipid in cells expressing the human GBA2 isoforms and control at 48 h after transfection, while the cluster maps illustrate the similarities of the patterns. The z-scores are color-coded from blue (lower than average for that lipid species) to red (higher than average for that lipid species), (**B**) ceramide (34:1, 36:1, 40:1, 40:2, 42:1, 42:2, and 42:3), (**C**) hexosylceramide (34:1, 40:1, 42:1, and 42:2), and (**D**) sphingomyelin (36:2, 40:1, 40:2, 41:1, 41:2, 42:1, 42:2, 42:3, 44:2, and 44:3) expressed as bar graphs. Values are means of three independent biological replicates with standard deviations shown as error bars, * *p* < 0.05 and ** *p* < 0.01 for differences compared to empty vector control in a two-tailed unpaired t-test. Shapiro-Wilk analysis showed no significant deviation from a normal distribution for those conditions showing significant differences.

**Figure 5 metabolites-10-00488-f005:**
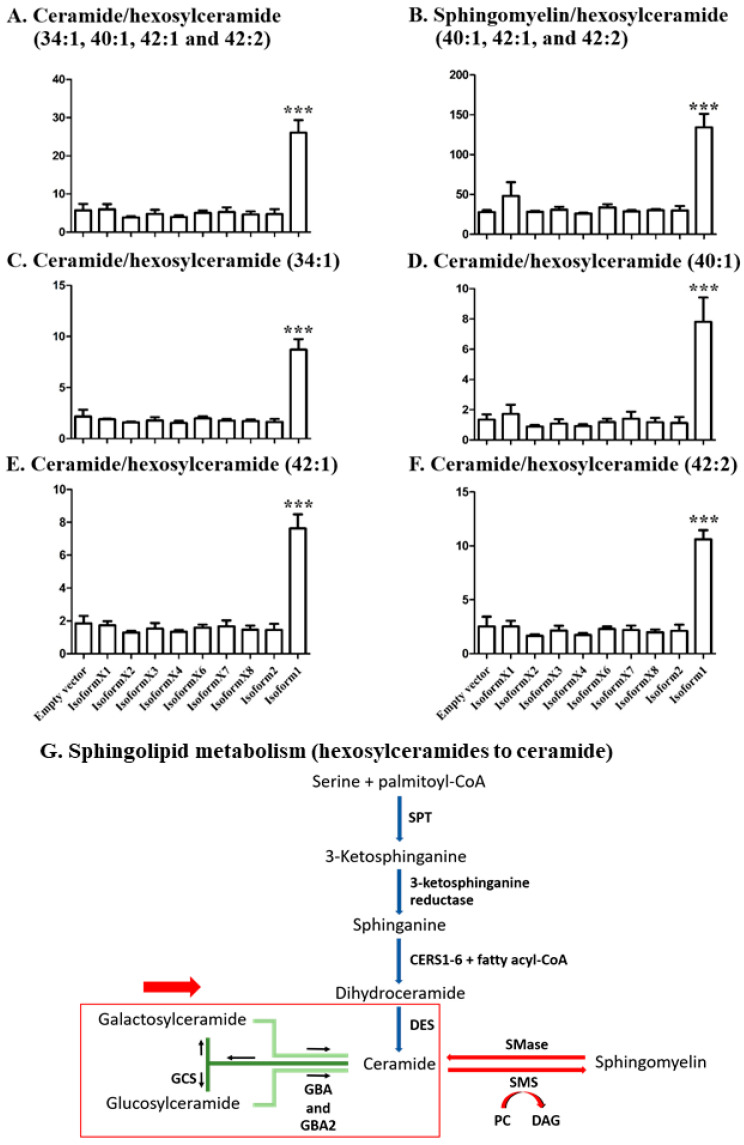
Effect of GBA2 isoforms on ceramide/hexosylceramide peak intensity ratios of specific ceramide species. (**A**) Ceramide/hexosylceramide (34:1, 40:1, 42:1, and 42:2), (**B**) sphingolyelin/hexosylceramide (40:1, 42:1, and 42:2) and ceramide/hexosylceramide ratios for 34:1, 40:1, 42:1, and 42:2 are shown separately in (**C**–**F**), respectively. Data are expressed as mean of three independent biological replicates ± SD, *** *p* < 0.01 significance for the difference from empty vector control in the unpaired t-test. Shapiro–Wilk analysis showed no significant deviance from a normal distribution for the samples with significant differences. (**G**) Map of sphingolipid metabolism showing hexosylceramide hydrolysis to release ceramides and their subsequent conversion to other species, which is affected by GBA2 overexpression.

**Figure 6 metabolites-10-00488-f006:**
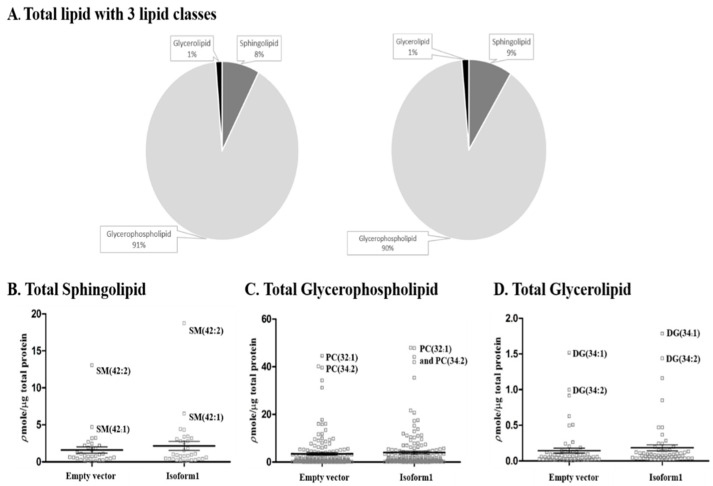
Relative levels of sphingolipids, glycerophospholipids, and glycerolipids in COS-7 cells transfected with empty vector and vector for human GBA2 isoform 1 for 48 h. (**A**) Relative amounts of three classes of lipid species. Levels of specific sphingolipid (**B**), glycerophospholipid (**C**), and glycerolipid (**D**) species and average values in control and cells expressing GBA2 isoform 1 are illustrated as parallel dot plots. Amounts were determined by mass spectrometry analysis of COS-7 cell lipid extracts. Results are representative of three independent replicates.

**Figure 7 metabolites-10-00488-f007:**
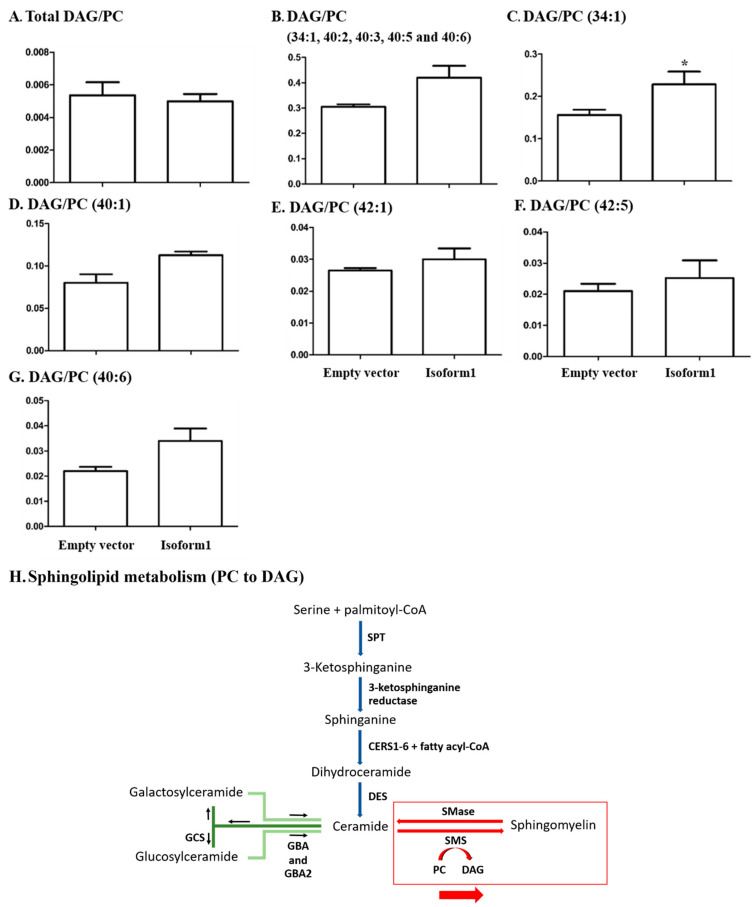
Ratios of levels of diacylglycerol to phosphatidylcholine in COS-7 cells transfected with control vector and GBA2 isoform 1 expression vector for 48 h. The ratios of total diacylglycerol (DAG) to total phosphatidylcholine (PC) are shown in (**A**). Total DAG/PC for those lipid species found in both lipid classes 34:1, 40:2, 40:3, 40:5, and d40:6) are shown in (**B**). The individual DAG/PC ratios for 34:1, 40:1, 42:1, and 42:2 are shown in (**C**–**G**), respectively. Data are expressed as mean of three independent biological replicates ± SD, * indicates *p* < 0.05 in the unpaired t-test. The distributions of sample values for all conditions did not deviate significantly from a normal distribution in a Shapiro–Wilk test. (**H**) Sphingolipid metabolism showing conversion of ceramide to sphingomyelin, which is affected by GBA2 overexpression, with emphasis on the conversion of PC to DAG during SM synthesis.

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
