# Peer review of "Effect of Expression of Human Glucosylceramidase 2 Isoforms on Lipid Profiles in COS-7 Cells"

_metabolites, 2020, doi:10.3390/metabo10120488_

Round 1

Reviewer 1 Report

I found this paper to be carefully put together and well written

It describes some quite detailed and logical work that I have no major concerns with re. methodology or conclusions drawn

My comments are therefore minor as below:

  • The studies referenced on page 3 – were there any population, or tissue location, specific effects beyond the localisation of isoform X1.
  • Figure 2D – May benefit from having either a split axis to show other isoform effects
  • Page 12 – Line 414 onwards. Possible discussion about the role of X1 if non-functional?
  • Page 12 – Line 431 onwards. Some discussion about why this effect is observed.
  • General discussion – Evidence that GBA2 expression impacts on function, discuss polymorphisms found in the general population/disease groups - in general a little more clinical correlation would lighten up a fairly technical manuscript
  • Page 15 – Conclusions. I think this would benefit from a little bit more context, I.e. link in disorders with dysregulated expression.
  • Clearer reference/methodology as to how the heat map and associated hierarchies were generated would be useful.

Author Response

  • The studies referenced on page 3 – were there any population, or tissue location, specific effects beyond the localisation of isoform X1. 
    Answer: To our knowledge, this is the first study to address the possible function of the different isoforms, other than isoform 1. In theory, some expression information might be extracted from the RNA seq data at NCBI, but they are short reads and could not confirm an entire isoform, just a particular splice site.
  • Figure 2D – May benefit from having either a split axis to show other isoform effects

Answer: We have split the axis as suggested to improve the graph.

  • Page 12 – Line 414 onwards. Possible discussion about the role of X1 if non-functional? Answer: We added some discussion of the possible effects of isoform X1 and other apparently non-functional isoforms in the second paragraph of the Discussion.
  • Page 12 – Line 431 onwards. Some discussion about why this effect is observed. Answer: We added a sentence to discuss this point: “Clearly, loss of GBA2 activity affects the glucosylceramide levels, which could also affect levels of more complex glycosphingolipids, although no significant decrease in ceramide was observed.”
  • General discussion – Evidence that GBA2 expression impacts on function, discuss polymorphisms found in the general population/disease groups - in general a little more clinical correlation would lighten up a fairly technical manuscript
    Answer: We have added a sentence describing some of the clinical phenotypes to the first paragraph of the Discussion:
    “The patients with these mutations have progressive diseases that can present from early childhood to early adulthood and have a wide range of neurological and non-neurological symptoms, including cataracts, hypotonia, brisk tendon reflexes, gait ataxia, spasticity, dystonia, and mental impairment, with thinning of the corpus callosum and atrophy of the cerebellum.” 
    In addition, we have corrected and expanded the last paragraph of the Discussion to include a recently described case of disease with GBA2 overexpression (in the model system at least) and the effects of the opposite case where GBA2 disrupting mutations lead to disease: “          A case of overexpression of GBA2 has been reported in the spinal cord of a superoxide dismutase mutant mouse model for amyotrophic lateral sclerosis (ALS), a fatal disease resulting in loss of motor neuron function (45).  Treatment with the GBA2 inhibitor ambroxol hydrochloride delayed loss of muscle strength and death in this model, and could also promote neuromuscular junction formation in tissue culture. GBA2 activity is increased by high substrate concentrations when lysosomal GBA is deficient in Gaucher and Niemann-Pick models (32), some symptoms of which are decreased when GBA2 is knocked out or inhibited (35, 37).  These symptoms were suggested to be caused by release of sphingosine in the cytoplasm, but our data suggest that changes in ceramide, glucosylceramide and DAG levels should also be considered. In contrast to these cases where knocking down GBA2 activity is desired, mutations that knock out GBA function lead to hereditary spastic paraplegia, autosomal recessive ataxia with spasticity and Marinesco-Sjogren-Like Syndrome (17, 18, 35, 36). Therefore, it is clear that the amount of GBA2 activity needs to be carefully controlled to allow proper maintenance of the central nervous system, as well as peripheral tissues.”
  • Page 15 – Conclusions. I think this would benefit from a little bit more context, I.e. link in disorders with dysregulated expression.
    Answer: To give the work more medical context, the following sentence was added to the end of the Conclusions: “The negative effects of loss of GBA2 function in HSP, ARCA and Marinesco-Sjögren-like syndrome, but positive effects of GBA2 inhibition in Gaucher, Niemann-Pick and ALS models, suggests that GBA2 is part of an important fine-tuning mechanism in lipid metabolism that is particularly critical in neuronal cells.”
  • Clearer reference/methodology as to how the heat map and associated hierarchies were generated would be useful.
    Answer: The heat map methodology has been expanded as follows:
    “The output data was visualized via GraphPad Prism 5.0 and R Core Team (2019; R: A language and environment for statistical computing.  R Foundation for Statistical Computing, Vienna, Austria. URL https://www.R-project.org/)  order script, setwd(‘’) to create the file directory, Lipid <- read.csv(‘file nmae.CSV', row.name = 1) to import the .CSV file to R, library("pheatmap") to download the library algorithm from the database, pheatmap(Lipid, cutree_rows = 4) to generate the heat map with clustering analysis s to group similarities in the heat map.  All values were expressed as Z-score (the position of a raw score in terms of its distance from the mean, z = (x-μ)/σ, where x is the mean value for samples of the same isoform, μ is the mean of the values of all isoforms and σ is the standard deviation between the means of all isoforms). For statistical analysis, the quantitative data were analyzed by 1-way ANOVA and Tukey's post-hoc test for multiple comparisons via GraphPad Prism 5.0 software. The mean differences were considered significant at p<0.05.”

Reviewer 2 Report

The authors identified GBA2 isoform 1 is the isoform that are active in cells, and showed us how GBA2 affect glycosphingolipid levels in cells. Overall, it is a good story. However, some minor things need to be improved.

1 Background in abstract should be terse. The more detailed background can be shown in Introduction section.

2 It's better to describe more why the authors chose COS-7 cells to do the test.

Author Response

1 Background in abstract should be terse. The more detailed background can be shown in Introduction section.

Answer: We agree. We eliminated 2 sentences of introduction from the abstract, although we added half a sentence to make the transition to the current work more smooth.

2 It's better to describe more why the authors chose COS-7 cells to do the test.

Answer: We have addressed the choice of COS-7 cells at the beginning of Section 2.2 as follows: “We chose COS-7 (Green monkey kidney) cells to evaluate the effects of the alternative splicing-generated sequence variations on GBA2 activity its effect on sphingolipids, because COS-7 cells have been shown to be an effective system for expression of GBA2 (22, 32). They were also reported to have undetectable levels of galactosylceramide (33), so that hexosylceramide levels should primarily reflect glucosylceramide.”

Reviewer 3 Report

In the manuscript, the authors tried to demostrat the functions and effects of human glucosylceramidase 2 (GBA2) variants in COS-7 vector expressing system, particularly they identified isoform 1 as the highest enzyme activity variant, while others showed no significant function. Over all, the study is well presented and results is interesting to readers. However, if possible, the authors may improve paper significance by analysing structural modifications among variants, though it was described in the Materials and Methods 4.7 SWISS-MODEL PyMOL, however, I didn't see results in the main figures.

Major concerns:

  1. Introduction section should be concise.
  2. Line 84, the variants should be presented by using IUPAC nomenclature. The current format is wrong.
  3. Statistical analysis should consult with experts. Please check carefully.

Author Response

In the manuscript, the authors tried to demostrat the functions and effects of human glucosylceramidase 2 (GBA2) variants in COS-7 vector expressing system, particularly they identified isoform 1 as the highest enzyme activity variant, while others showed no significant function. Over all, the study is well presented and results is interesting to readers. However, if possible, the authors may improve paper significance by analysing structural modifications among variants, though it was described in the Materials and Methods 4.7 SWISS-MODEL PyMOL, however, I didn't see results in the main figures.

Answer: Regarding the structural analysis, we had moved the structures to the supplementary figures compared to a previous version, which we agree was de-emphasizing it too much. We have brought the summary of the structural changes back to the main text in the new Figure 2.  We have also improved the comparison of the isoform structures in the supplementary figures (SI Figure 

Major concerns:

  1. Introduction section should be concise.

Answer: We shortened the Introduction by 20%, but we kept all the subjects covered since need to cover various aspects of GBA2 and sphingolipid metabolism to prepare the reader to evaluate the results.

  1. Line 84, the variants should be presented by using IUPAC nomenclature. The current format is wrong.

Answer: Yes, in line 84, the accession number to the reference structure was missing, but after adding it, we eliminated that example to make the Introduction shorter, as specified by the reviewer.  However, the comment also refers to the sequence variations caused by alternative splicing, so we corrected these to be in IUPAC format with the reference sequence version given at the beginning of the paragraph (second paragraph of the Results).  These sequence variation designations were also corrected in the structure figure.

  1. Statistical analysis should consult with experts. Please check carefully.

Answer: We consulted a statistician who said the methodology is adequate and it is similar to what was used in other metabolite papers from the group.  However, we improved the description to make this clearer:
“The output data was visualized via GraphPad Prism 5.0 and R Core Team (2019; R: A language and environment for statistical computing.  R Foundation for Statistical Computing, Vienna, Austria. URL https://www.R-project.org/)  order script, setwd(‘’) to create the file directory, Lipid <- read.csv(‘file nmae.CSV', row.name = 1) to import the .CSV file to R, library("pheatmap") to download the library algorithm from the database, pheatmap(Lipid, cutree_rows = 4) to generate the heat map with clustering analysis s to group similarities in the heat map.  All values were expressed as Z-score (the position of a raw score in terms of its distance from the mean, z = (x-μ)/σ, where x is the mean value for samples of the same isoform, μ is the mean of the values of all isoforms and σ is the standard deviation between the means of all isoforms). For statistical analysis, the quantitative data were analyzed by 1-way ANOVA and Tukey's post-hoc test for multiple comparisons via GraphPad Prism 5.0 software. The mean differences were considered significant at p<0.05.”

Round 2

Reviewer 1 Report

Thank you for responding to the comments constructively.

Reviewer 2 Report

The authors answer all my question. 

Reviewer 3 Report

Most of the comments have been appropriately answered, and the paper is much improved. However, for the statistics, I suggest the authors use "Nonparametric tests" for group case n (number) <30. In Figure 3, please indicate how many replicates in each group, including western blot and activity testings. If n<30, please use Kruskal–Wallis one-way analysis of variance. In Figure 4, the z scores are fine now. In Figure 5, please use the Kruskal–Wallis one-way analysis of variance for three independent replicates. In Figure 6, try the Mann–Whitney U test because only two tested groups are involved. These are the basic requirements. Please also refer to https://en.wikipedia.org/wiki/Nonparametric_statistics. If the statistician insists the methodology is adequate, please describe why the data fit normal distribution and why ANOVA is suitable for all these studies.

Author Response

We appreciate the reviewer's help with statistics. Please see the attached file for the response. 

Round 3

Reviewer 3 Report

I have no more comments